



# Predicting the steady-state isochronal stratigraphy of ice shelves using observations and modeling

Vjeran Višnjević[1], Reinhard Drews[1], Clemens Schannwell[2], Inka Koch[1], Steven Franke[3], Daniela Jansen[3], and Olaf Eisen[3,4]

[1]Department for Geoscience, University of Tübingen, Tübingen, Germany
[2]Max Planck Institute for Meteorology, Hamburg, Germany
[3]Alfred Wegener Institute, Helmholtz Centre for Polar and Marine Research, Bremerhaven, Germany
[4]University of Bremen, Bremen, Germany

**Correspondence:** Vjeran Višnjević (vjeran.visnjevic@uni-tuebingen.de)

**Abstract.** Ice shelves surrounding the Antarctic perimeter decelerate ice discharge towards the ocean through buttressing. Ice-shelf evolution and integrity depend on the local surface accumulation, basal melting and on the spatially variable ice-shelf viscosity. These parameters are often poorly constrained by observations and introduce uncertainties in ice-sheet projections for the ice-sheet evolution. Isochronal radar stratigraphy is an observational archive for the atmospheric, oceanographic and

ice-flow history of ice shelves with potential to assist model calibration. Here, we explore the possibility of using a simple and observationally driven ice-flow forward model to predict the ice-shelf stratigraphy for a given atmospheric- and oceanographic scenario. We validate this approach with the full Stokes ice-flow model Elmer/Ice and present a test case for the Roi Baudouin Ice Shelf (East Antarctica), where we compare model predictions with radar observations. The presented method enables us to investigate whether ice shelves are in steady-state and to delineate how much of the ice-shelf volume is determined by

its local surface accumulation. This can be used to better understand variability in ice-shelf rheology and for estimations which ice shelves are particularly susceptible to changes of surface accumulation rates in the future. Moreover, the numerically efficient prediction of isochronal stratigraphy is a step forward towards integrating radar data into ice-flow models using inverse methods. This has potential to constrain ocean-induced melting beneath Antarctic ice shelves using the ever-growing archive of radar observations of internal ice stratigraphy.

## 15  1  Introduction

The Antarctic Ice Sheet holds a sea-level equivalent of 58 m of global sea level rise (Fretwell et al., 2013; Morlighem et al., 2020) and some studies suggest a sea level contribution of up to 40 cm by the end of this century (Levermann et al., 2020; Edwards et al., 2021). Forming at the outlets of the ice/ocean boundary surrounding Antarctica, ice shelves play a major role in these future projections. The stability of ice shelves has been a focus of many studies (Rignot et al., 2008; Jansen et al., 2010;

Gudmundsson, 2013; Alley et al., 2016; Fürst et al., 2016; Banwell, 2017; Schannwell et al., 2018), and predicting their future behavior requires understanding the impact of future changes in atmospheric and oceanic forcing on their structural integrity. Uncertainties in ice shelf evolution are in part due to an insufficient understanding of the processes which drive ocean-induced





melting at the grounding-line and further seawards. The internal ice-shelf stratigraphy, as imaged by radar, is an underexplored archive which can be used to better constrain model predictions.

In grounded ice, the geometry of isochronal radar reflection horizons has been used in numerous studies in conjunction with ice-flow modeling to unravel ice dynamics (Nereson et al., 1998, 2000; Nereson and Waddington, 2002; Hindmarsh et al., 2009; Waddington et al., 2007; Catania et al., 2010; Leysinger Vieli et al., 2011; Lenaerts et al., 2014; Jenkins, 2016; Holschuh et al., 2017; Born and Robinson, 2021; Jouvet et al., 2020; Sutter et al., 2021), or surface accumulation history (Waddington et al., 2007; Lenaerts et al., 2014, 2019; Pratap et al., 2021) of various sectors in Antarctica or Greenland. This approach also gives insight into dynamic processes, e.g., such as basal sliding (Holschuh et al., 2017) and englacial folding (Bons et al., 2016;

Jansen et al., 2016). In ice shelves, similar concepts have been applied to derive the surface accumulation (e.g., Pratap et al., 2021) and basal melt rates (Pattyn et al., 2012; Matsuoka et al., 2012; Das et al., 2020). The internal radar stratigraphy also holds information with respect to the ice shelf's dynamic history and internal structure (Das et al., 2020), but this has been less often exploited so far.

In this study, we use ice shelf surface velocities derived by remote sensing to approximate a 3D velocity field of the ice shelf using a simple kinematic ice flow forward model. With this, we can predict the ice-shelf stratigraphy for a given set of oceanic and atmospheric boundary conditions and compare this to radar observations. One outcome is the delineation of two distinct ice bodies, namely ice that is formed from local accumulation seawards of the grounding line, and ice that is advected from upstream of the grounding line. Consistent with previous publications (Das et al., 2020), we refer to these two ice bodies as

continental meteoric ice (CMI) and local meteoric ice (LMI). The implications of the LMI/CMI ratio of a particular ice shelf are twofold: First, the two ice bodies can have different rheological properties because CMI, deposited further upstream, may contain colder and stiffer ice that protrudes into the ice shelves from their tributary ice streams (Larour et al., 2005; Khazendar et al., 2011). This ice may also be imprinted in terms of crystal orientation fabric from its source region (e.g., Alley, 1992; Thomas et al., 2021) whereas the LMI will not. Second, ice shelves that contain a large fraction of LMI are more susceptible to

changes in atmospheric precipitation compared to ice shelves which are predominantly sustained by their tributary ice streams. This is relevant for future ice-shelf stability because some predictions suggest a stable or even decreasing surface accumulation for ice shelves in contrast to the overall predicted increase for the rest of the Antarctic continent (Kittel et al., 2021).

## 2  Methods

The governing equation to predict the ice stratigraphy is the advective age equation:

$$\frac{\partial A}{\partial t} + \mathbf{v} \cdot \nabla A = 1. \tag{1}$$

The age of ice $A$ at depth $z$ is given by aging (i.e., the source term on the right-hand side) and ice advection, where $\mathbf{v}$ is the ice velocity $(V_x, V_y, V_z)$. Contours of the age field provide a natural comparison to the isochronal radar observations. The two main challenges in implementing this approach are (1) dealing with numerical diffusion, and (2) prescribing the

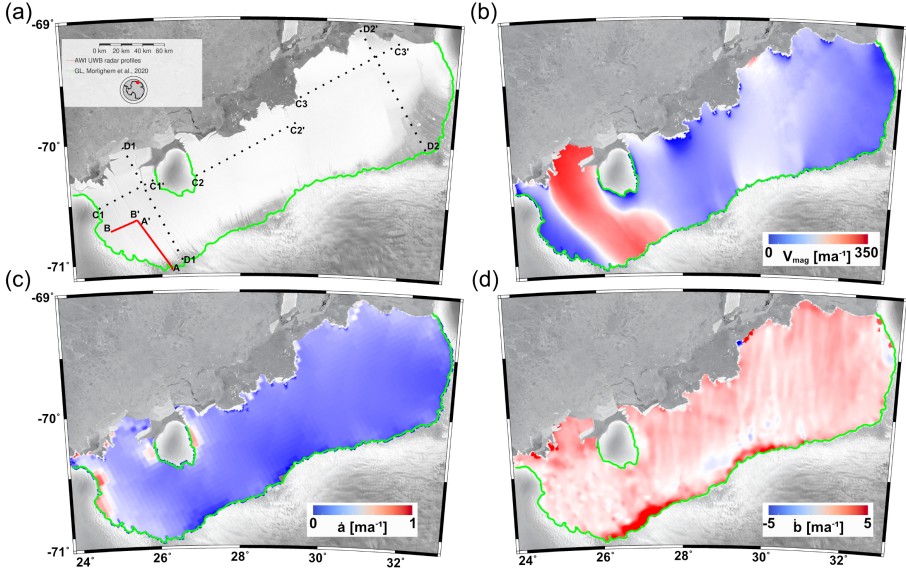

**Figure 1.** a) Overview of the Roi Baudouin Ice Shelf located in Dronning Maud Land, East Antarctica. The grounding line (Morlighem et al., 2020) is marked in green and radar profile lines, A-A' and B-B', used for validation are located in red. Black dotted lines C1, C2, and C3 correspond to the profiles shown in Fig. 6. The Radarsat Mosaic (Jezek, 2003) is shown in the background. (b) Surface speed (Gardner et al., 2018, 2019), (c) Surface accumulation rate (positive for mass gain, Lenaerts et al. (2014)), (d) Basal melting rate (positive for mass loss, Adusumilli et al., 2020).

three-dimensional velocity field. Numerical diffusion in classical discretization schemes can be minimized (e.g., Greve et al., 2002) but not fully avoided unless other approaches such as tracking the deformation in time and not space are implemented (Born, 2017; Born and Robinson, 2021). We will quantify the degree of diffusion by comparing the numerical predictions with analytical solutions for a specific test case (Sect. 2.3, 3.2). The velocities required for Eq. (1) are often modeled and as such include all the uncertainties typical for ice-flow modelling, e.g., uncertainties in the Glen flow index (Bons et al., 2016) or the ice softness parameter (Zeitz et al., 2020). We circumvent this step by assuming the flow regime of ice shelves where the horizontal velocities do not significantly change with depth (i.e., the shallow shelf assumption (SSA), Morland, 1987; MacAyeal, 1989; Weis et al., 1999). We illustrate this in the following sections.

## 2.1 Prediction of ice-shelf stratigraphy with observed surface velocities

We adopt a coordinate system such that $z = 0$ corresponds with the ice surface and $z$ increases with depth. The $x$- and $y$-directions correspond with the along- and across-flow direction, respectively. The horizontal velocities at the surface $(V_x, V_y)$ are obtained from observations and due to negligible basal friction these change little with depth. In steady state, this enables the analytical derivation of the vertical velocities (Drews et al., 2020) as a function of the local ice thickness $H$, the surface accumulation rate ($\dot{a}$, positive for mass gain), and the basal melt rate ($\dot{b}$, positive for mass loss):




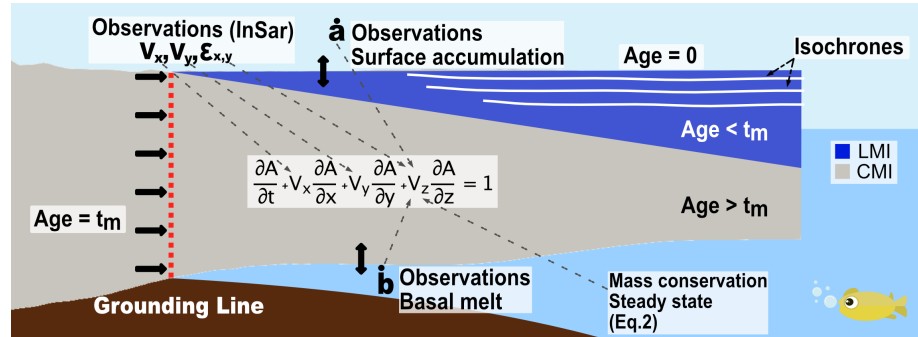

**Figure 2.** Scheme of the idealized ice shelf representing the distinction between the local meteoric ice (LMI, blue) and the continental meteoric ice advected from the ice sheet (CMI, gray). Isochrones are calculated by solving (Eq. 1), using observed horizontal velocities ($V_x$, $V_y$) and derived vertical velocities including fixed surface accumulation and basal melt rate fields (Eq. 2). On the inflow boundary upstream of the grounding line the age of ice is set to the length of the simulation time ($t_m$).

$$V_z = \frac{\rho_i}{\rho_w}\dot{a} + (1 - \frac{\rho_i}{\rho_w})\dot{b} + [H(1 - \frac{\rho_i}{\rho_w}) + z]\nabla \cdot \mathbf{V_H} \qquad (2)$$

where $\rho_i$ and $\rho_w$ are the respective densities of ice and water, and $V_H$ is a vector containing horizontal surface velocities.

With the three-dimensional velocity field at hand, Eq. (1) is solved numerically using the open source finite element library Elmer/Ice (Gagliardini et al., 2013). Velocity gradients are calculated using the strain rate solver in Elmer, and a separate solver is written to pass the vertical velocities from Eq. (2) to the age solver.

The age solver solves Eq. (1) with a semi-Lagrangian scheme (Martín and Gudmundsson, 2012). We set $A(z = 0) = 0$, assuming that there is no ablation at the ice surface (Fig 2). Due to a lack of observational age constraints at the grounding line,

the age is initialized with 0 seawards of the grounding line. Upstream thereof, the age is initialized with the simulation runtime $t_m$. These initial conditions provide an easy way to delineate the CMI-LMI boundary by finding the line $A = t_m$. Although convenient, such choice of initial conditions leads to a kink in the age-depth profiles at the contact point between LMI and CMI. This cannot adequately be captured with the semi-Lagrangian tracing scheme and deviations from the analytical and numerical schemes will be particularly evident around this transition (Sect 3.2).

The results are verified by comparing the predicted isochrone stratigraphy with radar-observed internal layering (A-A', B-B'; Fig. 1) which are typically considered to also be isochronous (e.g., Winter et al., 2019). Post-processing of the modeled age fields is done in Paraview (Henderson, 2004).

Steady-state conditions are assumed throughout. The ice-shelf geometry is obtained from BedMachine Antarctica (Morlighem et al., 2020), horizontal surface velocities are taken from satellite observations (Gardner et al., 2018, 2019), the surface accu-

mulation rate from RACMO 3.5 (Lenaerts et al., 2014), and the basal melt rate from a compilation of satellite derived surface height and velocity data combined with a firn layer model (Adusumilli et al., 2020). Those input data sets were chosen to





enable later an expansion of the methodology to all Antarctic ice shelves. For the test-case of the Roi Baudouin Ice Shelf, the continental products were cross-checked with the available higher resolution local products (Berger et al., 2017).

We run the simulations for 500 years, with 100 vertical layers, a horizontal resolution of 1km and a time step of 0.1 year. The

computation time for a single simulation using 100 CPUs on 5 nodes is 22 h on a cluster with Intel Xeon E5-2630 processors.

## 2.2 Validation of derived velocities with a synthetic 3D full Stokes model

Because Eq. (2) only holds in areas where the shallow-shelf assumption is valid, the analytical derivations are cross-checked in a synthetic case with an isothermal, isotropic, 2D, full Stokes model implemented in Elmer/Ice (Gagliardini et al., 2013). The model setup strictly follows the geometry and parameters used in the Stdn experiment in the marine ice-sheet modeling

intercomparison project (Pattyn et al., 2012), only with reduced ice shelf length (500 km), $\dot{a} = 0.3$ ma$^{-1}$ and $\dot{b} = 0$ ma$^{-1}$, and the grounding line condition set to first floating (Gagliardini et al., 2013). The model is initialized close to a steady state geometry and is run to a steady-state over 2500 years. Twenty vertical layers are used (corresponding to a mean spacing of around 20 m) and linearly increasing element spacing in the horizontal, ranging from less than 10 m near the grounding line up to 10 km towards the terminus. The modeled horizontal velocities are then used to derive vertical velocities using Eq. (2)

and are compared to the FS vertical velocity calculated by Elmer/Ice. This comparison naturally highlights areas such as the grounding zone where the SSA assumptions of depth-invariable horizontal flow are violated, and the differences compared to the FS solution are the largest.

## 2.3 Quantification of numerical diffusion

The degree of diffusion is quantified as a function of the number of vertical layers ($N_z$). Predictions are compared to an

analytical solution of A(x,z) and the linked LMI/CMI boundary. This is done by considering an unbuttressed, time-invariant ($\frac{\partial A}{\partial t} = 0$ in Eq. 1), two-dimensional ice shelf with constant and depth-invariant horizontal velocities. In that case the vertical velocity is also constant so that the trajectory along the LMI/CMI boundary $z_{DL}(x)$ of a particle deposited at the grounding line (GL) reduces to:

$$z_{DL}(x) = -\frac{V_z}{V_x}(x - x(GL)), \text{for } x > x(GL). \tag{3}$$

The corresponding age field in the LMI body increases linearly with depth:

$$A(x,z) = -\frac{z}{V_z}, \text{ for } z < z(LMI), \tag{4}$$

and is constant with depth for the CMI body:

$$A(x,z) = \frac{A_0 V_z + V_z\big(x - x(GL)\big)}{V_x V_z}, \text{ for } z > z(LMI). \tag{5}$$



**Table 1.** *Parameters used in 2D synthetic ice shelf test Section 2.3*

|  | Constants | Value | Units |
|---|---|---|---|
| $L_x$ | Length of Domain | 60 | km |
| $N_x$ | Number of points in x | 1269 | |
| $N_z$ | Number of vertical layers | 10/25/50/100/200 | |
| Time | Simulation time | 300 | a |
| $V_x$ | Horizontal Velocity | 200 | $\mathrm{ma}^{-1}$ |
| $V_z$ | Vertical Velocity | 1 | $\mathrm{ma}^{-1}$ |

The LMI/CMI boundary for more general velocity fields can also be obtained from streamline tracing and here this is done
with an implementation from Paraview (Fig. 4).

### 2.4 Radar observation and validation site of the Roi Baudouin Ice Shelf

The Roi Baudouin Ice Shelf is located in a comparatively narrow ice-shelf belt surrounding coastal Dronning Maud Land, East
Antarctica. This ice shelf is well suited to demonstrate the feasibility of our approach as previous studies in that area have
quantified the surface accumulation rates (Lenaerts et al., 2014) and basal melt rates (Berger et al., 2017) at comparatively high
resolution including ground-truth data. Moreover, analysis of the radar stratigraphy across ice rises (Drews, 2015; Callens et
al., 2016) in that area indicates that the entire catchment is likely to have been close to steady-state for the last several decades.
The ice shelf is dissected with numerous ice shelf channels which are located in areas in which our model assumptions of
depth-invariant horizontal velocities are likely violated (Drews, 2015; Drews et al., 2020; Wearing et al., 2021).

The radar data used in this study were collected with AWI's (Alfred Wegener Institute, Helmholtz Centre for Polar and
Marine Research) multichannel ultra-wideband (UWB) radar system (Rodriguez-Morales et al., 2014; Hale et al., 2016) in
the austral summer 2018/19 using the eight-element radar system which is installed on AWI's Polar 6 BT-67 aircraft. Data
were recorded in a frequency range of 150-520 MHz at an altitude of 365 m above ground. For radar data processing, which
comprises of pulse compression, synthetic aperture radar focusing, and array processing (for details, see Rodriguez-Morales
et al., 2014; Franke et al., 2020), we used the CReSIS Toolbox (CReSIS, 2020). The synthetic aperture radar processing was
optimised to increase the sensitivity of larger angle returns to achieve a better resolution of steeply inclined internal reflectors
(Franke et al., 2022). The final radargrams have a range resolution of 0.35 m and a trace spacing of 6 m.

Radar internal reflection horizons (IRHs) were traced semi-automatically using a 'maximum search'-tracking algorithm.
The traveltime-to-depth conversion uses a velocity of radio waves in ice of 1.68 m/ns and includes a firn-depth correction using
depth-density profiles from ice-cores collected in the area (Hubbard et al., 2013). The topographic correction and referencing
to sea level is done consistently with the model geometry obtained from BedMachine Antarctica (Morlighem et al., 2020). We
traced 8 internal reflection horizons (IRH1-IRH8) across the transect A-A' and 2 internal reflection horizons (IRH9-IRH10)
across the transect B-B' located in Figure 1a. Along the A-A' transect, we cannot trace IRHs until approximately 15 km
downstream of the grounding line. This is due to surface melt water infiltration near the grounding zone preventing formation

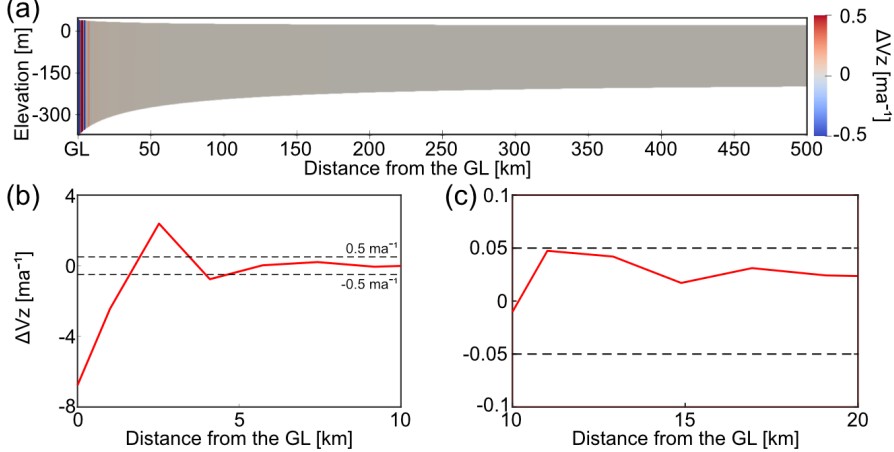

**Figure 3.** (a) Differences between the vertical velocity calculated using Eq. (2) and the full Stokes vertical velocity obtained by Elmer/Ice. (b) Transect values (red) from (a) for the first 10 km away from the GL and at 0 m elevation. Dashed black lines represent the $\pm\,0.5$ ma$^{-1}$ interval. (c) Transect values (red) from (a) between 10 km to 20 km away from the GL and at 0 m elevation. Dashed black lines represent the $\pm\,0.05$ ma$^{-1}$ interval.

of shallow layering (Lenaerts et al. 2017), and the CMI structure in this area where internal reflection horizons are also absent

in tributary Raghnhild Ice Stream (Callens et al., 2012). Therefore, we will compare the observations with the model results from the point where the first shallow IRHs start to emerge within the LMI on the ice shelf.

## 3 Results

In the following, we explore the limits of the approximated velocity field (Sect. 3.1) and quantify the degree of numerical diffusion of the age solution by comparing it to an analytical test case (Sect. 3.2). We then proceed by predicting the age

stratigraphy of the Roi Baudouin Ice Shelf and draw out numerous characteristics which are compared to radar observations (Sect. 3.3). We close by mapping the LMI/CMI boundary across the ice shelf.

### 3.1 Assessment of the analytical formula for vertical velocity

In the synthetic test case, the analytical approximation of the vertical velocities (Eq. 2) reproduces the full Stokes prediction with a mean deviation of 0.009 ma$^{-1}$ (corresponding to 3 % of the total vertical velocity) for distances > 10 km away from

the grounding line (Fig. 3c). Closer to the grounding-line the misfit increases with oscillating patterns reaching deviations of around $\pm\,0.76$ ma$^{-1}$ ($\sim\pm\,70$ %) at approximately 5 km distance (Fig. 3b).





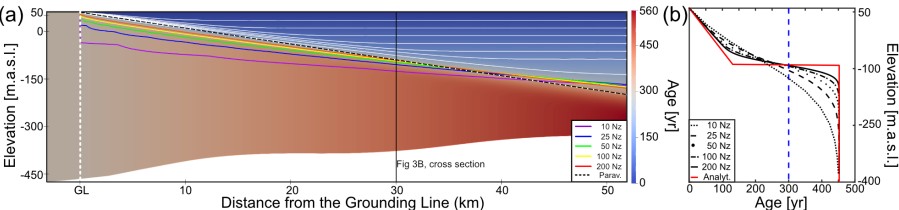

**Figure 4.** (a) Modeled Age field and the delineation line between locally accumulated ice and the advected ice for a different number of vertical layers, $N_z$ (Legend), analytically calculated delineation (white dots) and the stream tracer solution from Paraview (black dashes). (b) Vertical profile of the ice shelf age for different $N_z$ at 30 km away from the grounding line. The analytical solution is represented by the red line. The blue line represents the value of the age chosen for the contours in Fig. 4a.

## 3.2 Uncertainties in the predicted age-depth fields

Uncertainties due to numerical diffusion in the synthetic test case are highlighted in the horizontal direction using the LMI/CMI delineation line (Fig. 4a), and vertically at a cross-section near $x$=30 km (Fig. 4b). In both cases, the misfit between numerical

and analytical solution decreases with an increase in the number of vertical layers. The mean error in the position of the delineation line decreases from $\sim 50$ m for 10 vertical layers to $\sim$10 m for 200 vertical layers. Overall, the error is only minorly reduced for more than 100 vertical layers. The kink in the age-depth profile is, to a certain extent, smeared in an approximately 25 m interval with the age of ice being too high in the LMI, and too low in the CMI. Because of this symmetry, the deflection point of the predicted age-depth profile coincides with the LMI/CMI transition (vertical line in Fig. 4b). The extent of this

diffusive zone increases with increasing simulation runtime ($t_m$). It is therefore necessary to minimize the simulation runtime and here this is done by choosing $t_m$= 300 a which corresponds to the advection time from the grounding-line to the ice-shelf front. Choosing longer $t_m$ will not add more information.

Stream tracing for a fixed velocity field is not affected by the simulation runtime and reliably captures the analytical LMI/CMI transition. This will be used in the following real-world scenarios as an additional control.

## 3.3 Modeled 3D stratigraphy of the Roi Baudouin Ice Shelf


The modeled age fields along transects D1-D1' and D2-D2' (located in Fig. 1a) show significant differences in the position of the LMI/CMI boundary, and the volume, between these two ice masses in the western (D1-D1') and eastern (D2-D2') part of the ice shelf (Fig. 5). In the west, the D1-D1' transect mainly consists of CMI, and the position of the LMI/CMI boundary is shallow throughout the transect. We find the opposite conditions in the eastern part of the shelf, where the D2-D2' transect

is dominated by LMI, and the last 20 km leading to the terminus consists solely of LMI. This is also reflected in the deeper position of the LMI/CMI boundary throughout the transect as well as wider spacing between isochrones in the LMI.

Across-flow cross-sections C1-C1', C2-C2' and C3-C3' show large variations in both the modeled age and position of the LMI/CMI boundary across each section (Fig. 6). The modeled age pattern follows the horizontal velocity field: a shallow LMI/CMI boundary and correspondingly older ages in the CMI occur where horizontal velocities are large. Vice versa, a





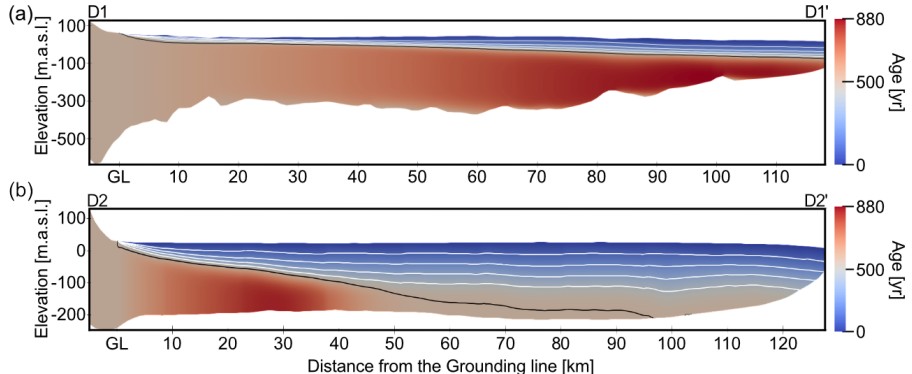

**Figure 5.** Modeled age profiles along (a) D1-D1', and (b) D2-D2' profiles shown in Fig. 1a. Direction of ice flow is from left to right. White lines depict 5 isochrones at constant intervals from the surface (age = 0) to age = 450 a. The bottom isochrone (black line) in each transect approximates to the LMI/CMI boundary (age = 500 a).

deeper LMI/CMI boundary and correspondingly older CMI ages occur where the ice flow is slow. Overall, the spatial extent of the LMI/CMI ratio varies greatly across the ice shelf with largest values in the eastern parts where some extensive parts are entirely sustained by LMI (Fig. 10). The western and the central parts of the ice shelf mainly consist of CMI. The inferred LMI/CMI boundary, solved using Eq.(1) are broadly consistent with the alternative approach of stream tracing from seed points located at the grounding line (Fig. 6).

In order to compare the modeled ice-shelf stratigraphy with observations, we visualize equally spaced isochrones from our predicted age field with the internal reflection horizons (IRHs) in the radar data along the transects A-A' (8 IRHs, Fig. 7) and B-B' (2 IRHs, Fig. 8). Cross-cutting of observed IRHs over modelled isochrones reflects imperfections in the model and in the applied boundary conditions. Because the age of the IRHs is unknown, we choose the predicted isochrones and observed IRH with similar mean depth for a one-to-one comparison (Nereson et al., 1998). Starting from the top, the predicted ages

of the radar reflection horizons along transect A-A' are: 28, 45, 73, 110, 140, 175, 230 and 305 years. Across transect B-B', perpendicular to the ice flow, the predicted IRH ages are 20 and 100 years. Furthermore, we interpolate the model age values along the observed IRHs, apply a moving mean, and plot the deviations of the age of each layer from its mean value (Fig. 9). Deviations from the mean accordingly highlight systematic misfits along the profile. Following the observed IRHs from the surface towards the bottom for A-A' (IRH1-IRH8) and B-B' transects (IRH9, IRH10), the standard age deviation for each IRH

equals: IRH1: 7, IRH2: 9, IRH3: 14, IRH4: 18, IRH5: 17, IRH6: 19, IRH7: 14, IRH8: 16, IRH9: 9 and IRH10: 17 years.

The misfits as exemplified by deviations from the mean age have systematic long (> 5km)- and short (< 5km) wavelength components. Close to the grounding line the predicted ages of IRHs (IRH1-IRH8) appear systematically too old, and the trend then reverses farther downstream (Fig. 9). Smaller scale deviations prominently correlate with, e.g., surface troughs in the topography. Such patterns are most likely indicative of an underresolution and systematic bias in the surface accumulation

rates, but other options are also possible (Sect 4.2).

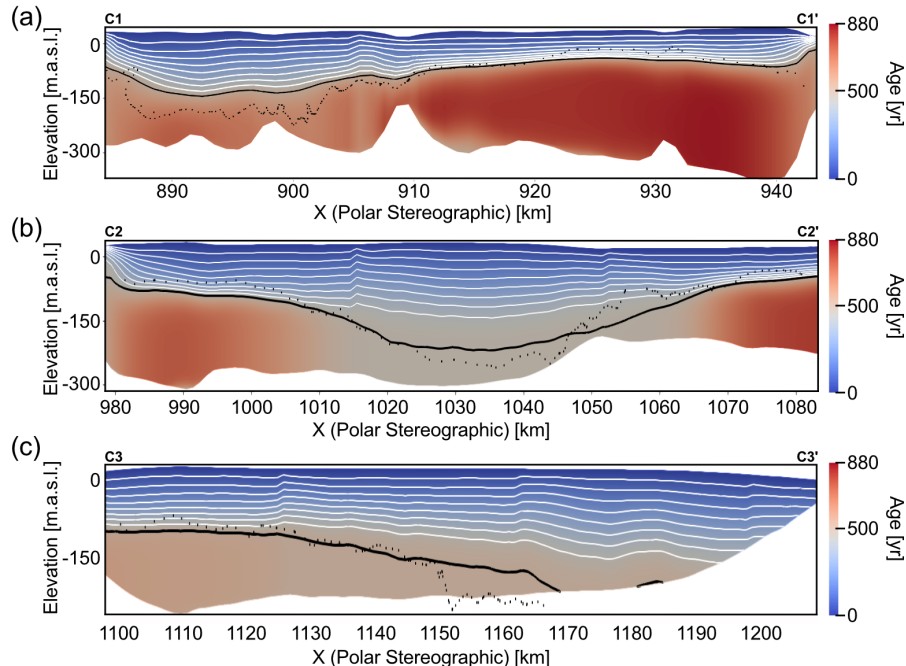

**Figure 6.** Modeled age profiles along (a) C1-C1', (b) C2-C2', and (c) C3-C3' transects shown in Fig. 1a. Direction of ice flow is into the image. White lines depict 10 contours at constant intervals from the surface (age = 0) to age = 450 a. The bottom isochrone (black line) in each transect approximates the LMI/CMI boundary (age = 500 a). Black dots correspond to the LMI/CMI boundary calculated using stream tracing.

## 4 Discussion

### 4.1 Advantages and shortcomings of the predicted age fields

Systematic mismatches between predicted stratigraphy and observations can be attributed to multiple reasons such as (1) numerical diffusion, (2) the applied boundary conditions, (3) violation of the shallow-shelf approximation, (4) violation of
the steady-state assumption, and (5) flawed surface accumulation or basal melt rate fields. Here, we will discuss these effects separately in the following.

Numerical diffusion, as shown in Fig. 4, is a consequence of the method used to solve the age equation (Eq. 1). Increasing the vertical resolution is the best way to counterbalance this effect, and here we found that 100 vertical layers ($N_z$) provide a good compromise between computation runtime and loss of accuracy. Importantly, the imprint of diffusion increases with
simulation runtime $t_m$, which is why $t_m$ should be set to the maximum characteristic time given by the ratio of length and average velocity of the respective ice shelf. The degree of diffusion for any simulation can be quantified by comparing the LMI/CMI boundary with stream tracing which is independent of the simulation runtime. Moreover, the age solver itself can

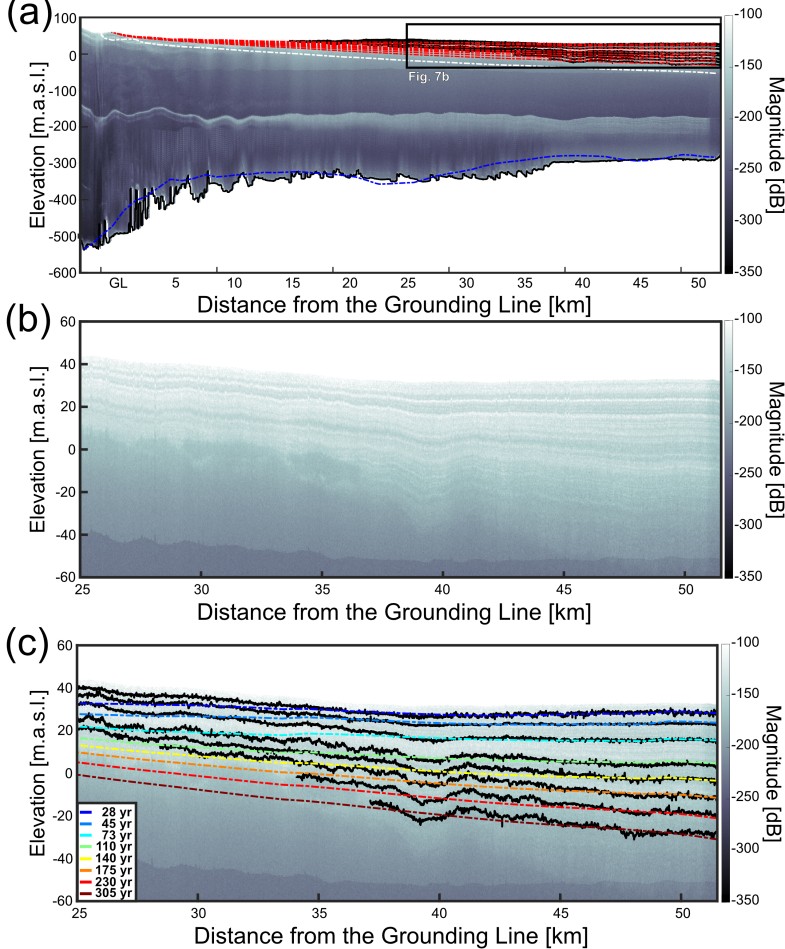

**Figure 7.** Comparison between radar observations and model output along A-A' transect (located in Fig. 1a). Direction of ice flow is from left to right. (a) Radar image of the modeled A-A' transect. Black lines represent digitized radar internal reflection horizons IRH1-IRH8, red lines represent modeled isochrones, and white dash-dotted line represents the LMI/CMI boundary. At the ice shelf/ocean boundary, the blue line is the ice shelf base used in the model (BedMachine Antarctica), and the black line is the ice shelf bottom picked from the radar image. Around 200 m of elevation, the prominent white reflection in the radar data is a multiple from the ice surface and bottom. (b) Zoomed in domain shown in the black rectangle from (a). Black lines are picked radar internal reflection horizons. Color dashed-dotted lines show modeled isochrones and their corresponding modeled ages (legend).

be replaced with other methods to solve the age equation (Born and Robinson, 2021), as the underlying host model for the velocities is simplistic and can be easily implemented in other frameworks.

The choice of a constant age value, $A(z, x = GL) = t_m$, at the inflow boundary currently precludes analysis of the radio stratigraphy in the CMI zone. In many examples, such as in our test case at RBIS or also in large parts of the McMurdo Ice Shelf (Das et al., 2020) this is acceptable, as the stratigraphy of inflowing tributary ice streams into the ice shelves is often



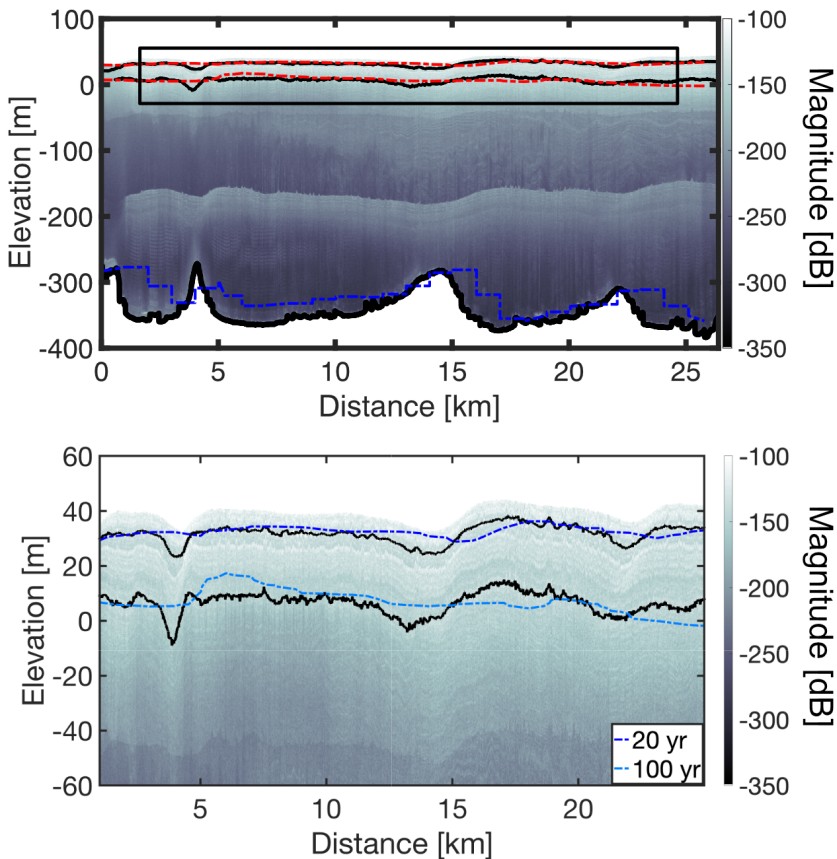

**Figure 8.** Comparison between radar observations and model output along B-B' transect (located in Fig. 1a). (a) Radar image of the modeled B-B' transect. Black lines represent picked radar internal reflection horizons IRH9 and IRH10, red lines represent modeled isochrones. At the ice shelf/ocean boundary, the blue line is the ice shelf base used in the model (BedMachine Antarctica), and the black line is the ice shelf bottom picked from the radar image. Around 200 m of depth, the white reflection in the radar data is a multiple reflection from the ice surface and bottom. (b) Zoomed in the domain shown in the black rectangle in (a). Black lines are picked radar internal reflection horizons. Dashed-dotted lines show modeled isochrones and their corresponding modeled ages (legend).

not well preserved. Nonetheless, in areas where slow-flowing ice enters the ice shelf, the stratigraphy in the CMI may well be intact so that the boundary condition applied is unsatisfactory. In principle, this can be solved by making the inflow boundary
condition one of the free parameters in the corresponding inverse problem.

The differences between the full Stokes Elmer/Ice vertical velocity and the analytical formula (Eq. 2) are small over the majority of the ice shelf, except within 5 km distance from the grounding line (Fig. 3), where more stress gradients are relevant than considered in the shallow shelf approximation (SSA). More specifically, this includes an oscillating pattern that emerges at the transition between the grounded ice sheet and the floating ice shelf (Lestringant, 1994; Durand et al., 2009). Ice-shelf





channel closure from lateral inflow is one example that is not adequately captured by the SSA (Drews, 2015; Wearing et al., 2021). Moreover, the surface accumulation rates likely change over these small spatial scales as discussed below. Also not included in this comparison are velocity modulations through ocean tides (e.g., Marsh et al., 2013; Rosier et al., 2017; Drews et al., 2021), although over the long timescales considered here this effect is likely to be negligible.

As the model domain starts at the grounding line, the incomplete kinematic model will predict an incorrect age profile for the shallow LMI at this location. This error will be maintained in the stratigraphy downstream. However, this effect is small because (1) the missed positive and negative oscillations will, to a certain extent, average out and (2) the ice residence time (typically several tens of years) across the grounding zone is an order of magnitude smaller than the characteristic time of ice shelves (typically many hundreds of years).

The steady-state assumption in Eq. (2) can, in principle, be extended to also include transient thickness changes as observed by satellite altimetry (Paolo et al., 2015), but the kinematic model does not handle the implied change in ice geometry. Therefore, an extension to transient cases requires more work. However, a systematic mismatch between the predicted and observed stratigraphy can be treated as a first order metric that transient changes have occurred in the first place.

Given that the vertical velocities in Eq. (2) are approximately nine times more sensitive to $\dot{a}$ than to $\dot{b}$ (Drews et al., 2020), it is imperative to have a well-constrained $\dot{a}$ field for reliable predictions of the isochronal stratigraphy. Errors in the $\dot{a}$ and $\dot{b}$ fields will propagate linearly into a misplaced LMI/CMI boundary. In future setups, the forward model presented here is a step towards an inverse approach to reconstruct $\dot{b}$ which is arguably least well known.

It is a logical next step to extend this method to other Antarctic ice shelves. The analysis of spatial variations of the LMI/CMI boundary across the ice shelves links to studies investigating ice shelf rheology and its spatial variations from inverse modeling (Larour et al., 2005; Khazendar et al., 2011). They find zones of stiffer ice and less stiff ice caused by the variability in ice temperature. In particular, ice from tributary ice streams is colder and stiffer. In our case, this will be mapped as CMI. Therefore, the spatial variations in the LMI/CMI boundary will likely reflect spatial variations of viscosity, implying that ice shelves with large LMI/CMI contrast also have a large contrast in viscosity.

Mapping and analyzing spatial variations in the LMI/CMI boundary can serve as a proxy for the sensitivity of the respective ice shelves to atmospheric or oceanographic perturbations (Levermann et al., 2020; Gilbert and Kittel, 2021). The buttressing capacity of ice shelves depends on their geometry (thickness) and structural integrity. The geometry of ice shelves dominated by LMI is hence controlled by the atmosphere, and consequently they are more susceptible to the projected changes in surface accumulation rates than ice shelves which are dominated by CMI. An atmospheric modeling study by Gilbert and Kittel (2021) suggests that the projected future increase in surface temperature nonlinearly increases the amount, duration and extent of surface melt and runoff, which increases vulnerability to hydrofracturing of ice shelves dominated by LMI. On the other hand, ice shelves dominated by the CMI will be more susceptible to the projected increase in basal melt (Naughten et al., 2018; Levermann et al., 2020; Gilbert and Kittel, 2021).

Furthermore, the predicted isochronal field can be tested against radar observations, particularly farther seawards, where the typically well-ordered stratigraphy of the LMI occupies a larger fraction. Systematic mismatches between observations and





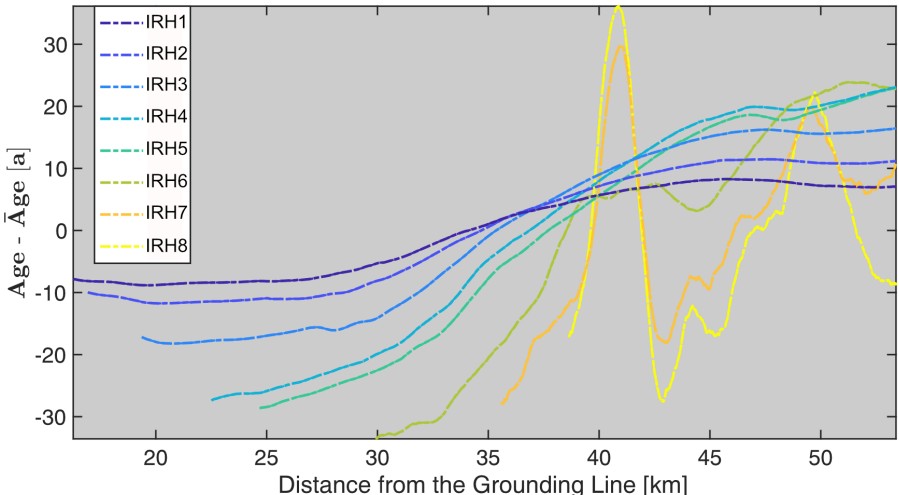

**Figure 9.** Modeled age deviations from the mean along picked internal reflection horizons from Figure 7, represented by lines IRH1-IRH8 starting from the surface towards the bottom.

predictions can then be discussed in terms of the forcing fields ($\dot{a}$, $\dot{b}$, $V_H$) and transient changes thereof. We further discuss
such a comparison for the Roi Baudouin Ice Shelf.

### 4.2  Model-data comparison at the Roi Baudouin Ice Shelf, East Antarctica

The across- and along-flow cross-sections shown in Figs. 5 and 6 present modeled expectations of the radar stratigraphy in the Roi Baudouin Ice Shelf for a given set of surface accumulation rate ($\dot{a}$) and basalt melt rate ($\dot{b}$) fields. These fields can give guidelines for the expected age-depth relationships when interpreting the stratigraphy from ice cores obtained from ice shelves
(e.g., Hubbard et al., 2013). They can also be used as a planning tool for radar surveys targeting the LMI where the stratigraphy is typically more intact than in the CMI.

The comparison of predicted isochrones with observations along A-A' captures the expected trend of a progressively increasing volume of the LMI, where the LMI/CMI boundary reaches a depth of approximately 40 m over a 50 km along-flow distance (Fig. 7a). In the radar profile, IRHs only develop approximately 15 km downstream of the grounding line, although surface
accumulation rates are positive throughout. Consequently, the model predicts formation of LMI everywhere. In this specific example, this can be explained with surface melt water infiltration in austral summers. Melt water forms at the surface due to ice-albedo feedbacks in a narrow belt near the grounding line (Lenaerts et al., 2017). This infiltration prevents the formation of a coherent radar stratigraphy even if the yearly averaged surface accumulation is positive.

Both the depth of the modeled isochrones (Figs. 7c and 8) and the age variations along the IRHs (Fig. 9) show that the mod-
eling result better reproduces IRHs closer to the surface. Further away from the grounding line, the model results overestimate the depth of the IRHs and do not reproduce the large scale trend until around the last 5 km for deeper to 15 km for shallow





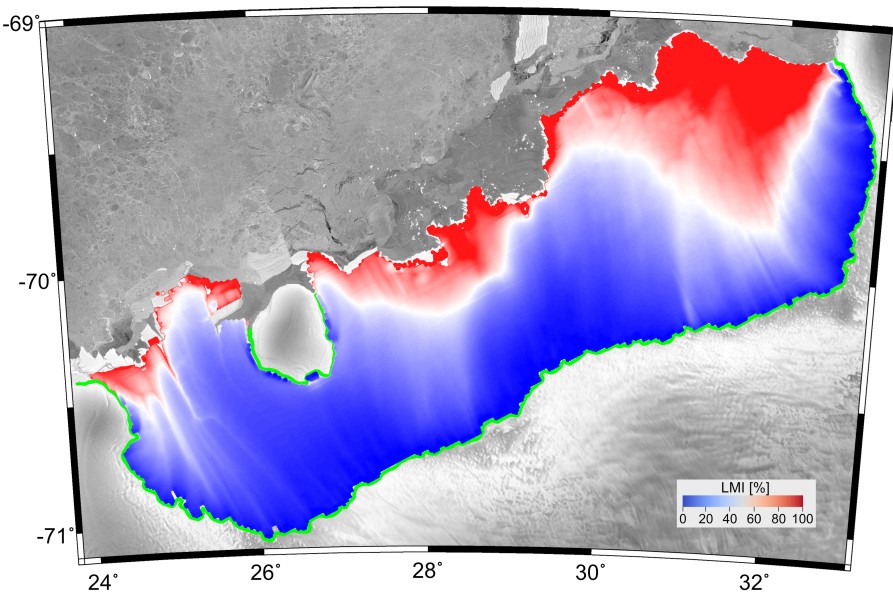

**Figure 10.** Percentage of local meteoric ice (LMI) over the ice shelf inferred from the steady-state age-depth fields. Green curve represents the position of the grounding line used in the model.

IRHs. Standard deviations between observations and predictions span between 7 to 19 years. Reasons for this are numerus and cannot be fully resolved here. It is possible that either the surface accumulation rates near the grounding line are too high, or that the basal melt rates are too low. For both cases (or a combination thereof), the misfit of predicted isochrones and IRHs
adds cumulatively over time and hence systematically increases with depth. Transient signatures in the applied fields are also a possibility and would need to be tested using inverse modeling. The same ambiguities also exist in principle for the smaller-scale variability around ice-shelf channels (Fig. 8). However, here it is most likely that this is caused by an under-resolving of processes in atmospheric precipitation which are known to sensitively depend on small-scale changes in surface slopes not resolved in the reanalysis data (Drews et al., 2020; Van Liefferinge et al., 2021).
Moving further away from the grounding line, we find the comparison of model predictions and observations on larger spatial scales encouraging, providing support that this particular catchment has not undergone large-scale changes in recent time. This is in line with other studies in this area that inferred similar statements in terms of ice-dynamic stability and linked atmospheric precipitation patterns on ice rises (Drews, 2015; Callens et al., 2016). Nonetheless, whether the presented misfit is dominated by local underestimation or potential small transient changes in model forcings (surface accumulation rate, basal
melt rate) is currently unclear and should be investigated in future studies.





### 4.3 Spatial variations in the percentage of locally accumulated ice

Mapped spatial variations show a non-uniform distribution in the percentage of LMI across the ice shelf (Figs. 5,6&10). The differences between the western and eastern parts are reflected in the surface velocity contrast between them (Fig. 1b). Unlike the western part, where the composition of the shelf is clearly dominated by CMI advected from the Western Ragnhild Glacier,
in the eastern part the total ice volume is dominated by LMI but the LMI/CMI ratio cannot be directly understood from the input data (Fig. 1).

This potentially makes the eastern part more likely to move out of close to steady-state conditions in the future, as its composition mainly depends on local surface accumulation - which is predicted to decrease (Kittel et al., 2021). Central and western areas of the shelf are less susceptible to these localized changes in mass input and output, as they mainly consist of
ice accumulated on and advected from the grounded ice sheet, where future projections for the end of this century project an increase in overall surface accumulation (Kittel et al., 2021).

### 5 Conclusions

The method here predicts the steady-state ice stratigraphy of an ice shelf using observed data (ice geometry, surface velocities, surface accumulation rate, basal melt rate) combined with a simple kinematic ice-flow model. In synthetic examples, we show
that the predicted vertical velocities correspond well to a full Stokes solution except in the first 5 km near the grounding line. Because the kinematic model is numerically efficient, it is possible to counterbalance numerical diffusion when solving the age equation with a high vertical resolution of 100 layers or more. Applications of this approach include calculation of the ratio between local meteoric ice (LMI) and continental meteoric ice (CMI) of ice shelves which serves as a first-order metric of the susceptibility of individual ice shelves to changes in atmospheric and oceanographic forcings. Moreover, comparing predictions
with observations from radar data provides a tool to estimate whether or not, for the atmospheric and oceanographic forcing from today, the stratigraphy which typically develops over many hundreds of years is in steady state. The numerical efficiency of the forward model may eventually also serve as a useful tool for the inversion of radar stratigraphy, e.g., with respect to the spatially variable basal melt rate fields.

The methodology is exemplified using the Roi Baudouin Ice Shelf in East Antarctica as a test case. The model-data mismatch
reduces moving away from the grounding line which could be due to unresolved variability in surface accumulation rates and basal melt rates in the input data for that area. The LMI/CMI ratio of this particular ice shelf varies strongly in the east-west direction and serves as a good example for two types of endmember ice shelves that are either primarily sustained by the local surface accumulation or by the ice flux of a tributary glacier. These will respond differently should atmospheric and oceanic circulation patterns change in the near future.



*Code availability.* Codes for the presented examples can be found at https://github.com/vjeranv/Visnjevic_et_al_2022_TC. Elmer/Ice version 8.4 (Rev: e6ab582) used is taken from https://github.com/ElmerCSC/elmerfem. Implemented Age solver is from Martín and Gudmundsson (2012).

*Author contributions.* VV performed all simulations and lead writing of the manuscript. RD gave input for the study design and model setup. CS advised simulations with Elmer/Ice and optimization of the super computing environment. IK analysed the airborne radar data which
were acquired by SF and DJ. All authors contributed to the writing/editing of the manuscript.

*Competing interests.* RD and OE are editors in The Cryosphere.

*Acknowledgements.* V. Višnjević, R. Drews and I. Koch were supported by an Emmy Noether Grant of the Deutsche Forschungsgemeinschaft (DR 822/3-1). CS was partially supported by the Deutsche Forschungsgemeinschaft (DFG) in the framework of the priority programme 1158 "Antarctic Research with comparative investigations in Arctic ice areas" by grant SCHA 2139/1-1. CS also acknowledges partial support by
the German Federal Ministry of Education and Research (BMBF) as a Research for Sustainability initiative (FONA) through the PalMod project. S. Franke was funded by the AWI Strategy fund, D. Jansen by the AWI Strategy fund, and the Helmholtz Young investigator group HGF YIG VH-NG-802. We thank the Kenn Borek crew, and Martin Gehrmann and Sebastian Spelz of AWI's technical staff of the research aircraft Polar 6 for support on the airborne radar survey. Logistical support in Antarctica was provided at Troll Station (Norway), Novolazarewskaja-Station (Russia), and Kohnen Station (Germany). Furthermore, we acknowledge using the CReSIS toolbox generated with
support from the University of Kansas, NASA Operation IceBridge grants NNX16AH54G, and NSF grants ACI-1443054, OPP-1739003, and IIS-1838230. The airborne radar data were acquired from the AWI within the CHIRP project.



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
