# Peer review of "Predicting the steady-state isochronal stratigraphy of ice shelves using observations and modeling"

_The Cryosphere, 2022_

## Author Comment (AC1)

*Dear Editors and Reviewers,*

*thank you for your positive feedback on our manuscript. We found the comments constructive and insightful. The main changes applied in the revised version pertain to (1) a clearer layout of our scientific motivation including an overhaul of the language, (2) clarification on some aspects in the methods, and (3) revisions of Figs 1, 2 & 7.*

*The conclusions remain unchanged. Below, all remarks are answered in detail. We found that the applied changes have improved the paper and hope that this is perceived by you in the same way.*

*On behalf of all coauthors,*

*Vjeran Višnjević*

RV1-1: Višnjevic et al. present a novel method to efficiently model the age structure of synthetic and Antarctic ice shelves and compare the latter to observed IRH's to deduce information on the strength and weaknesses of model assumptions and climate forcing with respect to mismatches against the observed internal stratigraphy. They further find that the Roi Baudouin ice shelf consists of two distinct regions which stability is either dominated by ocean or atmospheric conditions with potentially important implications for its future stability. The findings and methods presented in the manuscript are timely, interesting, and fit very well with the scope of The Cryosphere. The authors introduce novel approaches to glean insights from observed ice shelf stratigraphy and mention opportunities for formal inversion in the future. The manuscript is generally well written and the results intriguing. I would consider this manuscript for publication in The Cryosphere. However, to strengthen the main findings and to make for an easier read, some improvements should be considered for the structure and discussion of the results. Furthermore, the analysis could benefit from two additional experiments investigating the effect of changing climate forcing which I suggest further below (not strictly necessary but would strengthen the analysis and the scientific value of this study).

*Thank you for providing a constructive and motivating review. As detailed below, we have incorporated many of your suggestions hopefully leading to improvements.*

**General comments**

RV1-2: It is sometimes difficult to identify what the primary focus of this manuscript is supposed to be. While reading, to me it seemed that you mainly focus on the CMI/LMI transition and its role for future ice shelf stability. However, in the abstract, the foremost motivation seems to be model calibration -> "potential to assist model calibration". Relatively late in the manuscript you also touch on the impacts of the steady state assumption, boundary conditions and importantly the climate forcing.

But these points are only very quickly summarized. I think they would merit a more in-depth analysis. The subsection titles are a bit general, and it is sometimes difficult to pick up the main thread of the story. I assume your main motivation is to improve our understanding of ice flow, ice shelf stability and climate conditions by targeting the isochronal structure of ice shelves. However, throughout the manuscript this motivation is at times buried underneath methodological discussions. I think it would help if you could clearly provide the main scientific motivation of the different sections and distinguish them from the mere methodological aspects (which is of course important in its own right) of this study.

***Agreed. We concur that the scientific motivation of our study got overprinted by some technical aspects of our work. We now streamline this better. Mapping out the age-depth structure of ice shelves has multiple advantages: (1) It delineates the LMI/CMI boundary under contemporary ice-dynamic, atmospheric and oceanographic conditions. This helps us to map out different ice shelf types and their susceptibility to changes in ocean- or atmospheric forcing (ice shelves with a larger LMI will be more susceptible). The different ice shelf types may also differ in rheology as the thermal regime is either dominated by local snow accumulation or by ice advection from tributary ice streams; (2) It provides a tool to compare observed isochrones with steady-state predictions. A significant model-data mismatch can either reflect flaws in the model input (velocities, surface- and basal mass balance), or transient signatures that are not accounted for. Disentangling both possibilities needs to be done on a case-by-case basis; (3) We show that numerical diffusion of the Age equation is a serious problem for the fast-flowing ice shelves requiring high vertical resolution (or alternative tracking schemes). The revised version contains changes in that regard, particularly in the abstract and the introduction.***

RV1-3: L143-146 are a case in point here. Also, in the abstract you mention that the presented method enables the investigation whether ice shelves are in steady-state. What method are you referring to? The identification of the CMI/LMI transition or the use of equation 2 for establishing the vertical flow profile (Drews et al., 2020)? With numerically efficient prediction you probably refer to eq. 2 right? It is not always straightforward for the reader to appreciate the scientific goals of this study throughout the manuscript.

***Agreed and cf. RV1-2. We changed the sentence in the Abstract to make it clearer: "A large LMI to CMI ratio hereby marks ice shelves whose buttressing strength is more sensitive to changes in atmospheric precipitation patterns. Moreover, a mismatch between the steady-state predictions of the kinematic forward model and observations from radar can highlight inconsistencies in the atmospheric- and oceanographic input data, or be an indicator for a transient ice-shelf history not accounted for in the model."***

Introduction:

RV1-4: In the introduction you mention that you focus on the CMI/LMI boundary but do not mention that you also look at the effects of (1) numerical diffusion, (2) the applied boundary conditions, (3) violation of the shallow-shelf approximation, (4) violation of the steady-state assumption, and (5) flawed surface accumulation or basal melt rate fields as outline in line 199-200. This should be motivated here as well as one of the premises of using isochronal geometry in models is to identify issues with model setups, the computed velocity field, and uncertainties in the climate forcing.

***We have added this sentence to the Introduction section:" To do this, we use ice shelf surface velocities derived by remote sensing to approximate an equilibrium 3D velocity field of the ice shelf using a kinematic ice flow forward model. We verify this model using Elmer/Ice. Next, we investigate the influence of numerical diffusion on the modeled age field, and apply the method to the Roi Baudouin Ice Shel. We interpret differences between observations and predictions of flawed forcings (oceanic, climatic) and/or transient signatures."***

Methods:

RV1-5: From my point of view the methods section is a little confusing. I think a more intuitive approach would be to start by introducing the model setup, i.e., the ice shelf configuration of which you model the isochronal/age structure (1st idealized MISMIP ice shelf, 2nd RBIS) and why and then discuss how you derive the velocity field and compute the age (equations 1 and 2). Following that you can elaborate how you validate eq. 2 with Elmer/Ice in the MISMIP3D experiment and how you quantify numerical diffusion in the Numerical Diffusion experiment.

***We understand that starting with the age equation rather than with the geometries/velocities as done more commonly elsewhere can be confusing. However, in this paper the age equation is central and not, e.g., the force balance from which the velocities are usually derived. This is why we state the Age equation first and then derive the required quantities from there. In order to make this more appealing, we changed the introductory paragraphs of the methods to clarify this better. The references of the old intro paragraph were moved to section 4.1.***

***"In the following, we derive the required 3D velocities from the surface velocities assuming plug-flow (i.e., the shallow shelf assumption, Morland, 1987; MacAyeal, 1989; Weis et al., 1999, sect 2.1). We validate this assumption using the full Stokes model Elmer/Ice in a synthetic geometry (MISMIP Exp 1, sec 2.2). Numerical diffusion in the solution of Eq. 1 is quantified with another test case (NumDiff experiment) where an analytical solution exists (sec. 2.3). We then apply this method to the Roi Baudouin Ice Shelf (RBIS experiment) where radar data are available. "***

Results:

RV1-6: It is sometimes unclear whether you discuss the modelled isochrones based on eq. 2 or based on Elmer/Ice (also in the figures) and when in the synthetic test case. Please make this more explicit throughout. I suggest coming up with abbrevations e.g. RBOI, Stdn, Synth. It adds confusion that you call both the MISMIP3d setup and the small 2D ice shelf you use to estimate the effect of diffusion *synthetic*.

***Agreed, we have incorporated abbreviations (cf. RV 1-5), we named the abbreviations MISMIP EXP 1, NumDiff, and RBIS.***

Discussions/Conclusions:

RV1-7: Here I am missing a statement on future steps/plans to e.g. include transient climate conditions, non-equilibrium ice flow, coupling to the grounded parts of the ice sheet and how your methodology can be applied beyond the differentiation between CMI and LMI etc.

***Agreed, we added the following subsection to the Discussion section***

***"4.4 Future development***

***In future applications, the analysis of the spatial variability in the LMI/CMI pattern can be expanded to give insights into the susceptibility to future changes in the atmospheric and oceanographic conditions for all ice shelves around Antarctica. Secondly, mismatch between model output and data can be used to refine atmospheric precipitation and ocean-induced melting on the finer spatial scale provided by the radar observations.***

***Coupling with larger scale ice-sheet model can improve the inflow boundary condition so that also the CMI can be included in this approach."***

Specific comments:

RV1-8: L1 maybe "moderate" is more fitting than "decelerate"?

***Corrected.***

RV1-9: L3 I wouldn't necessarily call surface accumulation and basal melting parameters in this context, maybe "these components of ice shelf mass balance"

***Corrected.***

RV1-10: L3 more concise -> … introduce uncertainties in projections of ice-sheet evolution.

***Corrected.***

RV1-11: L16 … holds a sea level equivalent ice volume of ca. 58 m

***Corrected.***

RV1-12: L18: … with maximum estimates of up to …

*Corrected.*

RV1-13: L19 … play a major role in these future projections due to their buttressing effect on glacier flow (e.g. Fürst et al., 2016).

*Corrected.*

RV1-14: P2 L25/26 I would disagree, that the isochronal structure of Greenland or Antarctica has been used numerously in conjunction with ice flow modelling. I'd say, that the internal stratigraphy of the grounded ice sheet is very much underexplored as well. To my knowledge, so far there is no isochrone constrained model reconstruction or projection of ice sheet/glacier dynamics except for first efforts in that direction (e.g., Jouvet et al., 2020 (mountain glacier), Born et al 2021 (Greenland), Sutter et al., 2021 (Antarctica)). The references listed are either aimed at determining ice age (e.g., Nereson and Waddington, 2002) or reconstructing surface accumulation (e.g. Leysinger Vieli et al., 2011). I suggest providing a clearer differentiation with respect to the focus of the cited papers. Otherwise, the reader might surmise that for the grounded ice, using englacial layers is already a standard procedure in models which is probably not what you intend to convey here.

*Agreed, we changed the sentence into:" In grounded ice, numerous localized studies have used radar observations to infer surface and basal accumulation history (Nereson et al., 1998, 2000; Nereson and Waddington, 2002; Hindmarsh et al., 2009; Waddington et al., 2007; Catania et al., 2010; Leysinger Vieli et al., 2011; Lenaerts et al., 2014; Jenkins, 2016; Holschuh et al., 2017, Lenaerts et al., 2019; Pratap et al., 2021) of various sectors in Greenland or Antarctica. These approaches also gave insights into dynamic processes, e.g., such as basal sliding (Holschuh et al., 2017) and englacial folding (Bons et al., 2016; Jansen et al., 2016). Recent studies have focused on using the geometry of isochronal radar reflection horizons to reconstruct the ice dynamics of mountain glaciers (Jouvet et al., 2020), and also on a continental scale in Greenland (Born and Robinson, 2021) and Antarctica (Sutter et al., 2021).*

RV1-15: L35 … to approximate an equilibrium 3D velocity field …

*Corrected.*

RV1-16: L35 is this approach new or has it been used in previous studies (e.g. the ones you mention in the previous sentence)? If it is new, I would explicitly state this here. If not I would mention how you do things differently.

*Corrected, we emphasize this is a new approach.*

RV1-17: L36 I assume you can only predict the ice shelf stratigraphy which is not formed by the inflow from the grounded part of the ice sheet? If that's the case, please reformulate the sentence. In fact, you differentiate between CMI and LMI in the next sentence. Part of this could be moved from the introduction into the methods, while only mentioning in the introduction that you try to predict the ice shelf

stratigraphy. The caveats (CMI vs. LMI) would then be discussed in the Methods instead of the Introduction.

*Agreed, we changed the current sentence to:" We present a new approach to predict the stratigraphy of locally accumulated ice on the ice shelf for a given set of oceanic and atmospheric boundary conditions. The predictions are then compared to radar observations."*

RV1-18 L38 when you write 'one outcome' do you mean 'major finding'? It seems that the CMI/LMI delineation is the main focus here.

*Corrected.*

RV1-19: L59 not sure whether "circumvent" is the right term (also, what is meant by step?). In a real-world application the ice shelf also depends on the tributary glacier and the transition across the grounding line, lateral stresses etc. all these aspects would include the challenges you mention for typical ice flow modelling. By focusing solely on the floating part of the ice sheet you must come up with boundary conditions for the ice age introducing uncertainties close to the grounding line (as you outline later). So maybe be more specific here with what you mean by circumvent. Also, your main reason to model the stratigraphy of ice shelves is not because you forego uncertainties associated with modelling grounded ice but rather because it is interesting in its own right. I suggest rephrasing this sentence.

*Corrected, we removed the word circumvent. The new sentence is:"In the following, we derive the required 3D velocities from the surface velocities assuming plug-flow (i.e., the shallow shelf assumption (SSA), Morland, 1987; MacAyeal 1989; Weis et al., 1999; sect 2.1)."*

RV1-20: L61 You illustrate the plug-flow character of ice shelves in the following sections? Maybe reword and motivate the Method sections more explicitly.

*Corrected, see RV1-19.*

RV1-21: L63 I would start section 2.1 with an introductory sentence instead of jumping right into the coordinate system description. Something along the lines: "We derive the unknown vertical ice flow from observed surface velocities assuming steady state conditions (Drews et al., 2020) …" and then introduce eq. 2 and the specifics (parameters) and how you arrive at this analytical expression.

*Agreed, we added the introductory sentence and reorganized the first paragraph of the subsection.*

*"We analytically derive the unknown vertical ice flow from observed surface velocities (Drews et al., 2020). In steady state, this enables us to calculate vertical velocities as a function of the local ice thickness H, the surface strain rates, the surface accumulation rate ($\dot{a}$, positive for mass gain), and the basal melt rate ($\dot{b}$, positive for mass loss)"*

RV1-22: Figure 1: I suggest using a different color scale for surface flow and accumulation rate (continuous and not divergent). The divergent red-blue cmap used now gives the impression of a transition.

***Corrected.***

RV1-23: Figure 1 caption: Basal melting rate - > basal melt rates (negative for freezing …

***Corrected.***

RV1-24: Figure 2: scheme -> schematic/sketch/illustration. The many lines pointing to the age equation make this sketch a little busy. I would suggest dropping them. Font size can be smaller and not all in bold. Not necessary to write 'observations' every time. You can specify in the caption where the forcing comes from. Also, if I understand correctly in the idealized case (Stdn, MISMIP3D) forcing is uniform and not from observations. This is a bit misleading. I like the fish though.

***Agreed, we have removed the word observation and bold typeset in Figure 2, but left the lines pointing to the equation. We substituted the scheme with an illustration.***

RV1-25: L143 unclear what "limits of the approximated velocity field" mean, specify e.g., "explore the limitations of the approximated velocity field compared to a full stokes model"

***Corrected.***

RV1-26: L145 what do you mean by "draw out numerous characteristics". Please elaborate and specify.

***Agreed, we removed the expression and just state that:" We then proceed by predicting the age stratigraphy of the Roi Baudouin Ice Shelf and compare it to radar observations (Sect. 3.3)"***

RV1-27: l74-76. Maybe I am missing something here, but if I understand correctly, the initial age of the ice shelf is set to 0 while the upstream age (grounded ice) is set to the simulation's runtime? This means to the current model time (i.e. after 200 model years upstream age is 200 years) or the total runtime of the experiment? Maybe make this more verbose here. Also, I am missing a justification for this assumption. In reality, the age of the ice column at the grounding line depends on the depth of the ice (probably with a rather steep gradient) and the history of grounded ice flow. This is either unknown or highly uncertain if you use ice sheet model data from the grounded ice as an input. Wouldn't it make sense (as first approximation) to use the 3D velocity field of an ice sheet model (e.g. one with inversion of basal friction based on surface elevation/flow) to estimate the age input at the grounding line. It would be still quite wrong but less wrong than simply setting it to simulation runtime?  How does the assumption of a uniform age throughout the ice column influence your results?

*Agreed. This inflow boundary condition precludes using information from the CMI, and this is unsatisfactory but difficult to avoid at the moment (cf. L213). Assigning a linear profile will surely result in a more realistic profile, but it is unlikely to adequately capture the age-depth relationship at the grounding line. Choosing the inflow boundary condition, the way it is done here acknowledges that we cannot adequately predict age in the CMI, and it provides us with a convenient way to delineate the LMI/CMI boundary with simple thresholding.*

*We corrected "simulation runtime" into total runtime of the experiment.*

*We added the possibility of using a large-scale model output to set the inflow boundary: "In principle, this can be solved by making the inflow boundary condition one of the free parameters in the corresponding inverse problem, or use the output of a large-scale model as input for the inflow boundary of a nested ice shelf simulation."*

RV1-28: L80-82 If the real age structure of the ice shelf is unknown, how do you compare predicted and observed stratigraphy not knowing which layer corresponds to which age? Please elaborate.

*Since the real age structure is unknown, we take the model predicted isochrone closest (in mean depth) to the radar IRH. We changed the sentence to:"The results are verified by comparing the predicted isochrone stratigraphy with radar-observed internal layering (A-A', B-B'; Fig. 1). For comparison we choose the predicted isochrone which has the same mean depth as the observed one."*

RV1-29: L88 What do you mean by "cross-checked"? Please clarify? What was the results of this cross-check?

*This sentence has been entirely removed during the revisions.*

Figure 3 and section 2.2

RV1-30: L94 for those how do not know what the Stdn experiment is (not mentioned in Pattyn et al., 2012 as it is part of the MISMIP3D intercomparison experiments), a little more information is needed here.

*Agreed, we corrected the name of the experiment from the paper – EXP1, and added the exact subsection in Pattyn et al. (2012) where the experiment is described.*

RV1-31: L95 here I am confused again. Do you do the cross-check for the analytical case on the same model domain (Stdn MISMIP3D) as for Elmer/Ice. If so it might be good to make this clear. Otherwise, the uniform forcing of Elmer/Ice (a=0.3 ma-1, b=0 ma-1) are very different from the input data you use eq.2 (Figure 1). In that case, why don't you use the actual ice shelf geometry and boundary conditions in Elmer/Ice as well? I assume you do not use Elmer/Ice right away due to computational demands.

*Agreed, we refrained from using Elmer/Ice with real geometry in this example both due to computational demands, but also because it would be difficult to obtain the analytical solution for that geometry. We only use the MISMIP EXP 1 experiment to compare velocities, not layers. We model layers in the experiment testing the influence of numerical diffusion but on a different synthetic ice shelf geometry.*

*We emphasize the two different geometries in subsections discussing each experiment:*

*L109, Validation of vertical velocities, Sect 2.2:*

*„The model is initialized close to a steady state geometry from the MISMIP EXP1 experiment (Fig. 3a, or see Section 3.1, Pattyn et al., 2012} and is run to a steady-state over 2500 years. "*

*L118-119, Quantifying diffusion, Sect 2.3*

*This is done by considering an unbuttressed, time-invariant, two-dimensional synthetic ice shelf (shown in Fig. 4a) with constant and depth-invariant horizontal velocities."*

RV1-32: L147 Subection 3.1 is only two sentences long. Either be more expansive here or consider merging subsections 3.1 + 3.2.

*We did not include this change; we would like to keep these subsections separated as they describe different experiments with different setups. We slightly extended the section, see RV1-33.*

Figure 3:

RV1-33: Wouldn't it be more interesting to show the difference in vertical velocity at two or more vertical profiles through the ice shelf (e.g. one 'borehole' at 10 km and one at 50 etc.). I assume the difference in the vertical velocity between Elmer/Ice and Eq.2 at the surface will be minimal by definition? This would also be more consistent with the data you show in 4b.

*No change in Figure applied. We stayed with visualizing the velocity difference along a horizontal slice because there is more variability along the horizontal than vertical axis by at least an order of magnitude. This way we clearly show at what distance moving away from the GL the difference between the velocities strongly reduces. We added to the document: "Variability along the vertical axis is less than 1% throughout."*

RV1-34: What significance has the pm 0.5 ma-1 delineation in Figure 3?

*We added the dashed lines to make it easier for the reader to understand the figure and more clearly see the error values. The chosen value itself is not significant, it's chosen so it represents the error range in the 5-10 km area.*

Figure 4:

RV1-35: Caption: It probably should read Fig 4b cross section on the vertical line in 4a (also check use of lower/upper case for consistency).

**Corrected.**

RV1-36: I do not see the white dots (calculated delineation). What are the white lines? I assume isochrones as in Figure 2?

***White lines are age isochrones in the LMI, added to the figure caption. The analytical solution overlaps with the one from Paraview so the white dots are between the black dashed lines. We emphasize this in the figure caption:***

***"NumDiff experiment (Sect 3.3) (a) Modeled Age field and the delineation line between locally accumulated ice and the advected ice for a different number of vertical layers, $N_z$ (Legend), stream tracer solution from Paraview (black dashes), and analytically calculated delineation (white dots -between black dashes)"***

RV1-37: Minorly is rarely used (I had to look it up actually)? Maybe change to slightly?

***Corrected.***

RV1-38: L177-179 difficult for me to follow, I suggest rephrasing this. Smeared? 25 m vertically I presume? Also depends on resolution, maybe provide a range for the resolutions tested here. E.g. 10-50 m or whatever would apply here.

***We reworded the sentence:" The kink in the age-depth profile is, to a certain extent, diffused over an approximately 25 m depth interval with the age of ice being too high in the LMI, and too low in the CMI (Fig. 4b)".***

***We added the resolution range in a sentence above:" The mean error in the position of the delineation line decreases from ~ 50 m for 10 vertical layers (vertical grid size of around 40 m) to ~10 m for 200 vertical layers (vertical grid size of around 2 m)."***

RV1-39: Figure 5: What's the reason for the bulge of relatively old ice between 10-40 km downstream of the grounding line? Looking at Figure 1 there seems to be no salient anomalous melt/refreeze or velocity variations along this transect? Maybe worth discussing this? In Figure 6b it seems clear that the old ice pockets correlate to the areas with relatively fast ice flow. But what happens here?

***The profiles in Figure 5 are taken along straight lines, and therefore do not fully follow the flowline. For that reason, the old ice pocket is just advected ice from the sides of the profile where the ice is slower and older.***

***We added this sentence at line 187:***

*"The transects D1-D1', D2-D2' are along straight lines, not fully following the flowline (Fig. 1a). Older ages of ice occurring in the CMI along that transect originate from the slower flowing margins containing older ice."*

RV1-40: L183-184 there is no reason to believe that the closest modelled isochrone should have the same age as the observed IRH, except if you are very confident about your velocity field. It is very probably that the modelled isochrone is either too young or too old, potentially by a lot. I am aware that it is unfortunately not possible to use ice core tie points or trace the nearest dated continental isochrone all the way to the ice shelf. But I think at this point there should be a discussion of the caveats of your comparison. If you discuss the elevation misfits of two isochrones of very different age it is not straightforward to assign these misfits to local mass balance/ice flow I guess. So, the question would be to what extent additional information can be extracted here except for the fact that cross-cutting is bad?

*No reason is a bit strong b/c the input fields are physically motivated. However, clearly this approach is only sensitive to the difference between surface accumulation and basal melting. Multiplying the input fields by a constant will result in the same isochronal stratigraphy (but with a different age that we cannot capture as you correctly point out). Therefore, as long as the age of the isochrones is unknown, we can strictly speaking only infer spatial patterns (and not their magnitudes). In that sense we now suggest more explicitly in the new section 4.4 that our method allows calibration/downscaling the spatial patterns of the Antarctic wide surface accumulation/basal melting. This is important, given that both of these fields sensitively depend on the topography which changes strongly on scales finer than current grid size of atmospheric forcing, and isochrones can be a tool to achieve this refinement.*

RV1-41: L186-188 I do not get this step. Do you mean by interpolating that you take the modelled isochrone above and below the IRH and then vertically interpolate the IRH's age? And then take the deviations between the nearest observed and modelled isochrone as a measure of systematic misfit? The temporal resolution of your isochrones is one year?

*Our model output is the age field not isochrones. We get isochrones by extracting them from the age field as contours of the same age. The temporal resolution corresponds to the time step applied (here 0.1 years).*

*We emphasize now that our we start from the age field and interpolate along the position of the IRHs:" Isochrones are obtained through contouring the finite-element age field. The predicted isochrone that is closest to the observed one in mean depth is used for comparison. Ages are also extracted at the coordinates of the observed isochrones, and the model-data misfit then corresponds to the demeaned differences acknowledging that the age of the observed isochrone is unknown."*

RV1-42: Subsection 4.1 title is a bit vague. Advantages and disadvantages with respect to what? I suggest being more explicit here.

*We changed it into: "Analysis of the age stratigraphy predicted by the kinematic ice flow model"*

RV1-43: L210 This or a similar sentence should also appear earlier when you discuss the methods.

*Agreed, see RV1-5.*

RV1-44: Figure 7: I think you use jet-cmap or similar, for me it's fine, but I think it is not a good choice for color-blind readers.

*Corrected. We added the following to the figure caption:*

*"Dashed-dotted lines (presented in the same order as in the legend) show predicted isochrone geometry and absolute ages (legend)."*

RV1-45: L200 "In the following we discuss these effects separately"

*Corrected.*

RV1-46: L230-231 please elaborate and specify what kind of model developments are necessary to make this work in your opinion. I assume the main issue is that forward models would diverge to much from the observed real evolution of an ice shelf.

*Agreed and as already indicated in the other changes above (cf RV1-7) we tone down the extension to transient cases.*

*We changed the sentence into:"Therefore, an extension to transient cases requires more work - especially on constraining transient changes in ice geometry and velocities, as well as in atmospheric and oceanic forcings.*

RV1-47: L233-236 please write out melt/freezing rate and accumulation."

*Corrected.*

RV1-48: If the maximum age of e.g. the RBIS is ca. 900 years old (according to your velocity fields) then past variations in ice shelf mass balance will probably be significant. You could test this by e.g. repeating your experiments with an mass balance forcing (melting/freezing and surface accumulation) which is e.g. half or twice (or maybe more realistically 10-20%) as large as the currently observed and have a look at how the age field changes. This would be a straightforward test to gauge the relative importance of time varying forcing.

*We have performed these experiments for small changes in SMB (+/- 15%) but do not include them in the manuscript. Due to the linear nature of Eqs. (1) and (2), increase or reduction in SMB will linearly increase or reduce the LMI volume. This holds for small changes in SMB, as larger changes in SMB will change the ice geometry. This is now explicitly included in the revisions (RV1-51).*

RV1-49: Figure 9 maybe more illustrative to use percent changes to the interpolated IRH age. A difference of 10 years for an IRH of age 10 is more serious than for an 800 year old IRH.

*We slightly disagree. We would prefer to leave this showing the age offset from the mean in years and not percentages. The difference of 10% for a younger shallow layer (around 10-20 years) is significantly smaller than for a deeper older layer (around 300 years), to avoid this confusion we kept the values in years.*

RV1-50: Figure 10: again, please use a continuous cmap. The diverging cmap insinuates a transition which is randomly set to 50% LMI. Ignore this statement if this effect is intended (but then please explain why a 50 % LMI delineation is relevant.)

*It is intended, the 50% line clearly distinguishes between areas dominated by the CMI and the ones dominated by the LMI. We add the following to the figure caption:*
*"The white contour marks the LMI/CMI ratio of 0.5. Areas marked in red are dominated by CMI, areas in blue by LMI. The green curve represents the position of the grounding line used in the model."*

RV1-51: L283-285 again, you could test this by scaling your input fields and discussing the changes in isochrone geometry according to your local climate drivers.

*Eq. (1) and (2) are both linear, therefore changes in their amplitudes will linearly change the age field, proportionally increasing the volume of the LMI.*

*We added:" Future changes, but also errors, in the accumulation and melt rate fields will propagate linearly into changes in the position of the LMI/CMI boundary (Eqs. 1&2)."*

RV1-52: L293-296 I would argue that this only holds for moderate changes in ice shelf mass balance. If surface and ocean warming are large enough than the current portion of CMI of an ice shelf probably does not matter much for its stability? If I recall correctly Kittel et al. show substantial increase in surface melt over ice shelves for the 21[st] century for high emission scenarios (e.g. for example if forced by CESM2). In some runs shown by Kittel also the grounded part of the ice sheet experiences widespread surface melt.

*Kittel et al. (2021) mention a potential warming threshold, +7.5˚C, which would lead to lower grounded SMB but they state that this needs to be confirmed in future studies. We decided here not to discuss the end member result but focus on the results of the +1.5˚C and +2.5˚C scenarios.*

*We changed the sentence:" Central and western areas of the shelf are less susceptible to these localized changes in mass input and output, as they mainly consist of ice accumulated on and advected from the grounded ice sheet, where future projections for the end of this century project an increase*

*in overall surface accumulation for the +1.5˚C and +2.5˚C scenarios (Kittel et al., 2021)."*

RV1-53: L310 reduces -> decreases

*Corrected*

---

## Author Comment (AC2)

*Dear Editors and Reviewers,*

*thank you for your positive feedback on our manuscript. We found the comments constructive and insightful. The main changes applied in the revised version pertain to (1) a clearer layout of our scientific motivation including an overhaul of the language, (2) clarification on some aspects in the methods, and (3) revisions of Figs 1, 2 & 7.*

*The conclusions remain unchanged. Below, all remarks are answered in detail. We found that the applied changes have improved the paper and hope that this is perceived by you in the same way.*

*On behalf of all coauthors,*

*Vjeran Višnjević*

RV2-1: Summary: This work uses a simple, observationally driven ice flow model to forward model ice shelf stratigraphy with a given atmospheric and ocean scenario. The method is validated with Elmer/Ice model. The model predictions are then compared with radar observations over Roi Baudouin ice shelf. The internal layers in the LMI region resolved by the radar are compared with the model-derived internal layers. As the ice shelf model uses a steady state assumption, this is a way to predict if an ice shelf is in steady state if the mode predictions agree with observations of internal stratigraphy.

*Thank you for your review. We have implemented many of the suggested changes detailed below.*

Major comments:

RV2-2: This is an important concept. The transition of LMI and CMI and the percentage of the ice shelves that comprise of LMI/CMI component can have important consequences for ice shelf stability. However, the paper is not very clearly written, in my opinion. The authors need to take another look at the sections to improve readability.

*Thanks for pointing this out. We implemented a number of structural changes also mentioned by RV1 including revision of abstract, introduction and methods (RV1 3-6 among others).*

RV2-3: The white LMI/CMI boundary in Figure 7- I assume it is modeled. Is there a possibility to provide an uncertainty in the depth of the LMI/CMI layer?

*In the initial and the revised versions, we quantify aspects of the uncertainty (cf. RV1-38) pertaining to the LMI/CMI boundary including one (1) numerical diffusion (Sect. 3.2), and (2) now more explicitly flaws in the input forcing (they propagate linearly into the calculation of vertical velocities (Eq 2) and in return*

*the calculated age (Eq 1)). Compared to those major points, other uncertainties (e.g., wrongly observed surface velocities or geometries) are minor.*

*We added a sentence:*
*" Future changes, but also errors, in the accumulation and melt rate fields will propagate linearly into changes in the position of the LMI/CMI boundary (Eqs. 1&2)"*

RV2-4: Why only Roi-Baudouin was chosen for validation? Was any other ice shelf considered for the validation with airborne radar? Is Ross Ice Shelf not a good candidate for comparison?

*We focus on presenting our methodology, show synthetic experiments, and apply the method to one real world ice shelf. In future work we plan to apply this method to all ice shelves around Antarctica, and will the also provide the climatic and oceanographic context of the individual ice shelves.*

RV2-5: The presence of marine ice may be sporadic on some ice shelves, but extensive on others (example the Ronne-Filchner ice shelf). The limitation of this method needs to be acknowledged, particularly along lines 234-240.

*This seems like a misunderstanding. The presence of marine ice will ideally be included in the basal melt rate fields and the dynamic effect will be included in the surface velocities. Therefore, there is no fundamental problem with and as Eq (2) in this case.*

RV2-6: Figure 10 needs a panel of basal melt rates from Adusumilli et al. The comparison of LMI/CMI composition with basal melt rates will be interesting and important. For example, does the basal melting pattern differ considerably on either side of the ice shelf?

*Such a figure was already included in the initial version. Basal melt rates, surface accumulation rates and velocities are displayed in Figure 1. In the Discussion section (Sect 4.3, L307) we state that the recovered LMI/CMI pattern mainly reflects the velocity field.*

Minor comments:

RV2-7: How do the upstream grounded ice surface look like in Figure 7? Are there crevasses, blue ice etc. that would prevent the identification of layers below the LMI/CMI interface? A figure delineating the possible surface conditions would be helpful.

*Stokes et al. (2019) report a widespread distribution of supraglacial lakes, as well as blue ice (Matsuoka et al., 2018), in the area upstream of the ice shelf.*

*Other studies also report on the lack of internal layering. Callens et al. (2012) report that there is no internal layering on the tributary Ragnhild Glacier.*

***We added:" In this specific example, this can be explained with surface melt water infiltration in austral summers as well as with the existence of supraglacial lakes in the area upstream of the ice shelf (Stokes et al., 2019)."***

References:

Matsuoka, K., Skoglund, A., & Roth, G. (2018). Quantarctica [Data set]. Norwegian Polar Institute. https://doi.org/10.21334/npolar.2018.8516e961

Stokes, C. R., Sanderson, J. E., Miles, B. W. J., Jamieson, S. S. R., & Leeson, A. A. (2019). Widespread distribution of supraglacial lakes around the margin of the East Antarctic Ice Sheet. Scientific Reports, 9(1), 13823. https://doi.org/10.1038/s41598-019-50343-5